# Increasing Chocolate’s Sugar Content Enhances Its Psychoactive Effects and Intake

**DOI:** 10.3390/nu11030596

**Published:** 2019-03-12

**Authors:** Shanon L. Casperson, Lisa Lanza, Eram Albajri, Jennifer A. Nasser

**Affiliations:** 1USDA, Agricultural Research Service, Grand Forks Human Nutrition Research Center, 2420 2nd Ave. North, Grand Forks, ND 58203-9034, USA; 2College of Nursing and Health Professions, Drexel University, 1601 Cherry St MS31030 RM 389, Philadelphia, PA 19102-1320, USA; lisa.marcelalanza.lanza@drexel.edu (L.L.); eram.abdullah.albajri@drexel.edu (E.A.)

**Keywords:** chocolate, Addiction Research Center Inventory, sugar, craving, addictive-like eating, eating behavior

## Abstract

Chocolate elicits unique brain activity compared to other foods, activating similar brain regions and neurobiological substrates with potentially similar psychoactive effects as substances of abuse. We sought to determine the relationship between chocolate with varying combinations of its main constituents (sugar, cocoa, and fat) and its psychoactive effects. Participants consumed 5 g of a commercially available chocolate with increasing amounts of sugar (90% cocoa, 85% cocoa, 70% cocoa, and milk chocolates). After each chocolate sample, participants completed the Psychoactive Effects Questionnaire (PEQ). The PEQ consists of questions taken from the Morphine-Benzedrine Group (MBG), Morphine (M,) and Excitement (E) subscales of the Addiction Research Center Inventory. After all testing procedures, participants completed the Binge Eating Scale (BES) while left alone and allowed to eat as much as they wanted of each of the different chocolates. We found a measurable psychoactive dose–effect relationship with each incremental increase in the chocolate’s sugar content. The total number of positive responses and the number of positive responses on the E subscale began increasing after tasting the 90% cocoa chocolate, whereas the number of positive responses on the MBG and M subscales began increasing after tasting the 85% cocoa chocolate sample. We did not find a correlation between BES scores and the total amount of chocolate consumed or self-reported scores on the PEQ. These results suggest that each incremental increase in chocolate’s sugar content enhances its psychoactive effects. These results extend our understanding of chocolate’s appeal and unique ability to prompt an addictive-like eating response.

## 1. Introduction

Chocolate holds a special status in our society. Indeed, it is one of the most loved and craved, but problematic, foods [1]. Consuming chocolate evokes pleasant feelings, reduces tension, and improves mood [2,3]. Furthermore, chocolate elicits unique brain activity compared to other high-sugar and high-fat foods, recruiting brain structures that respond to craving-inducing stimuli, and is therefore more likely to provoke an addictive-like eating response [4]. The particular combination of cocoa, sugar, and fat in chocolate may play important, yet distinct, roles in chocolate’s unique ability to elicit an addictive-like eating response. Smit et al. [5] demonstrated a role of the main psychopharmacological active constituents of cocoa in producing psychostimulant effects but determined that other attributes, such as sweetness and texture, may be more important. In a prior study, we observed effects of the percent cocoa and sugar contents on “desire to consume more chocolate”, while fat content trended towards significance for this effect [6]. Defeliceantonio et al. [7] demonstrated a supra-additive effect of combining sugar and fat on food reward in humans, while others have demonstrated that the sugar component (in a sugar/fat combination) in particular is more effective at activating reward [8,9] and gustatory brain circuits [9]. Additionally, we have demonstrated that the highly reinforcing properties of sugar are difficult to overcome [10,11]. Taken together, chocolate’s desirability appears to arise from the synergistic relationship among its components.

Fat and sugar are known to stimulate both the dopamine and the opioid neurotransmitter systems to regulate a food’s rewarding potential [12,13,14]. The dopamine neurotransmitter system stimulates ‘wanting’, or the motivation to consume the food [12,14], whereas the opioid neurotransmitter system modulates the consumption of a desired food by amplifying ‘liking’, or the hedonic value, of the desired food [12,13]. Thus, the sight, smell, and taste of a highly palatable food such as chocolate work together to trigger motivational and hedonic reward mechanisms that result in the pursuit and consumption of the desired food. Subjective dopaminergic and opioidergic effects of food consumption can be differentiated using the Addiction Research Center Inventory (ACRI) [15]. We previously utilized the ARCI, specifically the Morphine-Benzedrine Group (MBG), Morphine (M), and Excitement (E) subscales, in a between-group design (groups identified by percent cocoa of the sample tasted), to provide indices of the psychoactive effects of chocolate that are associated with addictive-like eating [6]. We showed that simply tasting chocolate increases the number of positive responses on the MBG subscale of the ARCI, consistent with responses obtained on the MBG subscale after dopaminergic–opioidergic drug administration [16]. These results help explain chocolate’s high reinforcing value, as foods that elevate feelings of euphoria have a greater reinforcing potential [17]. However, different individuals may specifically crave a particular combination of chocolate’s main components (dark vs milk chocolate).

To continue our study of the psychoactive effects of chocolates varying in percent cocoa and sugar and fat content, we repeated our 2011 study using a within-subject design. Because some have postulated that “food addiction” is more of an “eating behavior addiction” similar to binge eating [18,19], we added the Binge Eating Scale (BES) [20] to our data collection. We hypothesized that self-reported scores on the MBG and E subscales of the ARCI would correlate with the particular combination of the main components (cocoa, sugar, and fat) of the chocolate and the amount of each of the different chocolates consumed after the tasting session. Consistent with our 2011 study, we expected no correlation between self-reported scores on the M subscale of the ARCI and the particular combination of chocolate’s main components. In addition, we hypothesized that the amount of chocolate consumed after the tasting session would positively correlate with the increasing sugar content and decreasing cocoa and fat content of the chocolate. Furthermore, we hypothesized that BES scores would positively correlate with self-reported scores on the ARCI questionnaire and chocolate consumption.

## 2. Materials and Methods

### 2.1. Participants

Healthy adults (Table 1) were recruited from the greater Grand Forks, ND, and Philadelphia, PA, areas. Our participant population consisted of 57% non-overweight, 30% overweight, and 13% class 1 obese participants. Screening for study eligibility included height, weight, and a medical health history questionnaire. Exclusion criteria included: presence of food and non-food allergies; current status as a dieter; current or past metabolic illnesses (diabetes, renal failure, thyroid illness, hypertension); psychiatric, neurological, or eating disorders (schizophrenia, depression, Parkinson’s Disease, Huntington’s Disease, cerebral palsy, stroke, epilepsy, anorexia nervosa, or bulimia nervosa); taking prescription medications except for oral contraceptives or antihyperlipidemia agents. The study was approved by both the University of North Dakota and the Drexel University Institutional Review Boards and registered on clinicaltrials.gov as NCT03364413. Informed written consent was obtained from all participants prior to any study-related procedures.

### 2.2. Experimental Procedures

Participants reported to the Center at which they were recruited 3–4 h postprandial to determine the psychoactive effect of consuming chocolate with varying amounts of fat, sugar, and cocoa (Table 2). Participants were instructed to have a light meal (e.g., sandwich with side salad or cup of soup) and to then refrain from eating or drinking (except water) before reporting for their study visit. Upon arrival, participants rated their appetite using 10 cm visual analog scales and completed a baseline Psychophysical Effects Questionnaire (PEQ; based on the ARCI) [6]. Participants were then presented with 5 g of commercially available chocolate varying in cocoa, sugar, and fat concentrations (Table 2). Chocolates were tested in order from least to most amount of sugar, and the PEQ was completed immediately after each chocolate tasting. Participants completed the Binge Eating Scale (BES) after the tasting session. Participants at the Grand Forks site were then left alone and allowed to eat as much as they wanted of each of the different chocolates.

### 2.3. Questionnaires

Subjective homeostatic and hedonic hunger ratings were assessed using 100 mm visual analogue scales. Questions asked included: 1. How hungry do you feel; 2. How strong is your desire to eat; 3. How full do you feel; 4. How satisfied do you feel; 5. How much do you think you could eat right now; 6. Would you like to eat something sweet; and 7. Would you like to eat something fatty.

The PEQ is composed of 30 questions taken from the ARCI Morphine-Benzedrine Group (MBG), Morphine (M), and Excitement (E) subscales that assess subjective dopaminergic and opioidergic effects of psychoactive drugs [15]. The MBG subscale contains questions that center on feelings of well-being and euphoria, which correlate with the activation of both the dopaminergic and the opioidergic neurotransmitter systems. The M subscale focuses on attitude and physical sensations, and the E subscale relates to physical and psychological feelings of excitement, both of which correlate with activation of the dopaminergic neurotransmitter system [16]. Participants completed the PEQ before and after tasting each of the different chocolates.

The BES is composed of 16 questions used to assess the presence of binge eating behavior [20]. The BES contains questions that assess both behavioral manifestations of binge eating and of the feelings that either cue or follow a binge episode. Participants completed the BES after completing all testing procedures.

### 2.4. Anthropometric Measurements

Height, measured to the nearest 0.1 cm using a stadiometer (SECA Model 214, Hamburg, Germany), and body weight, measured using a calibrated digital scale to the nearest 0.1 kg (Fairbanks model 50735; Kansas City, MO, USA), were obtained at the end of the testing session.

### 2.5. Statistical Analysis

General linear models were used to compare general participant characteristics (i.e., sex, age, body mass index (BMI), hunger, testing site) and ARCI subscale scores of each chocolate type (defined by the percent cocoa in the chocolate sample tasted) and the amount of chocolate consumed. Tukey’s post hoc analysis was used to determine differences. The threshold of significance was set at alpha = 0.05. JMP V14 (SAS Institute, Inc., Cary, NC, USA) was used for all analyses.

## 3. Results

### 3.1. Subjective Appetite Responses

On a scale from 0 (not at all) to 100 (very much), participants’ rating of hunger was 58 ± 21 and of the desire to eat was 66 ± 28, while fullness was 25 ± 24 and feeling of being satisfied was 31 ± 21. There was no main effect of hunger or fullness on the total number of positive responses and on the number of positive responses on the MBG and M subscales; however, there was a main effect of hunger (F(1,15) = 4.73, *p* = 0.046) and fullness (F(1,15) = 8.06, *p* = 0.012) on self-reported scores on the E subscale of the PEQ.

Participants also rated their desire to eat foods with a specific taste profile on a scale from 0 (not at all) to 100 (very much). Participants’ rating of the desire to eat something sweet was 79 ± 14 and that of the desire to eat something fatty was 48 ± 27. There was no main effect of the desire to eat something sweet or fatty on the total number of positive responses or on the number of positive responses on the MBG, M, and E subscales.

### 3.2. Psychophysical Effects Questionnaire

There was no main effect of testing site, sex, age, BMI, or BES on the total number of positive responses or on the number of positive responses on the MBG, M, and E subscales. Because of the direct correlation between the percent cocoa and sugar contents and the simultaneous changes of the fat and cocoa contents, we were not able to assess the direct effects of the individual chocolate components in this within-subject study design.

There was a significant main effect of chocolate type on the total number of positive responses (F(4,106) = 30.10, *p* < 0.0001) and on the number of positive responses on the MBG (F(4,106) = 15.83, *p* < 0.0001), M (F(4,106) = 10.34, *p* < 0.0001) and E (F(4,106) = 16.57, *p* < 0.0001) subscales. Tukey’s post hoc analysis revealed slight differences in the effect of chocolate type on the total number of positive responses and on the number of positive responses on MBG, M, and E subscales.

The total number of positive responses (Figure 1) significantly increased after the consumption of all the different types of chocolate, with milk chocolate eliciting the greatest increase.

The number of positive responses on the MBG subscale (Figure 2) significantly increased after the consumption of the 85% cocoa chocolate and continued to increase in a dose-dependent manner in response to tasting each of the other chocolates. The number of positive responses on the MBG subscale after the consumption of the milk chocolate was significantly greater than after consumption of any of the other chocolates.

The number of positive responses on the M subscale (Figure 3) did not increase after the consumption of the 90% cocoa chocolate. A significant increase above baseline was not observed until participants consumed the 85% cocoa chocolate, with no further increases for each sequential chocolate consumption.

The number of positive responses on the E subscale (Figure 4) significantly increased after the consumption of the 90% cocoa chocolate. As with the MBG subscale, the number of positive responses on the E subscale continued to increase in a dose-dependent manner in response to each incremental increase in the chocolate’s sugar content and decrease in the percent cocoa and fat content.

### 3.3. Chocolate Consumption

There was a main effect of chocolate type on the amount of chocolate consumed (*p* < 0.0001). Overall, participants consumed significantly more milk chocolate (21 g ± 25 (SD)) than any other chocolate type (90% cocoa: 3 g ± 9; 85% cocoa: 2 g ± 4; 70% cocoa: 6 g ± 10 (SD)). There was no correlation between the amount of chocolate consumed and the total number of positive responses or the number of positive responses on the MBG, M, or E subscales. There was a main effect of hunger (F(1,18) = 11.74, *p* = 0.003) on the total amount of chocolate consumed. Post hoc analysis revealed a main effect of hunger (F(1,17) = 8.64, *p* = 0.009) on the amount of milk chocolate consumed only. There was no main effect of BES score on the total amount of chocolate consumed.

## 4. Discussion

The current study aimed to extend our previous findings on the psychoactive effects of consuming chocolate varying in cocoa, sugar, and fat concentrations using a validated ARCI “drug effects” questionnaire in a within-subject design. The questions used simultaneously reflect alterations in motivation, mood, sensation, and perception, and, therefore, provide insight into the interrelation of these variables and chocolate consumption [16]. Our data indicate a measurable psychoactive dose–effect relationship with each incremental increase in the chocolate’s sugar content and decrease in the percent cocoa and fat contents. Overall, there were an inverse dose–effect relationship with cocoa concentration and fat content and a positive dose–effect relationship with the sugar content of a chocolate. In addition, our data indicate that the dose–effect relationship of the different chocolates was slightly different for each ARCI subscale. Thus, the present study is the first to demonstrate a dose-dependent relationship between self-reported scores on the PEQ and chocolate consumption.

Increased feelings of well-being, euphoria, and physical and psychological feelings of excitement after chocolate consumption are consistent with chocolate’s ability to modulate both the opioid and the dopamine neurotransmitter systems. Both human and animal research has demonstrated the reinforcing potential and comforting and mood-ameliorating effects of chocolate [2,3,21,22,23]. Contrary to our hypothesis, we did not find an association between self-reported scores on the MBG subscale and chocolate consumption. These results are also inconsistent with our previous study in which self-reported scores on the MBG subscale were associated with an increased desire to eat more chocolate [6]. A possible explanation for these findings is that subjective appetite measurements were obtained prior to the tasting session rather than after; however, participants were allowed to consume as much of the different chocolates as they wanted after the tasting session. On average, participants consumed 8 ± 15 g more chocolate than what was provided to them for the tasting session. Another possible explanation for this difference is that the smell and tasting of four different chocolates in the same session as opposed to only one type of chocolate could have satiated the desire to eat more chocolate. Massolt et al. [24] demonstrated that not only eating but simply smelling chocolate (85% cocoa) suppresses appetite. Additionally, Sørensen and Astrup [25] demonstrated that dark chocolate (70% cocoa) increases satiety and decreases the desire to eat something sweet more than milk chocolate. In the current study, participants consumed 15 g of dark chocolate before tasting the milk chocolate, and this could have decreased their appetite for more chocolate.

The sugar content, which plays a key role in chocolate’s pleasurable taste and texture, is important in determining chocolate’s reinforcing potential [26]. Research has shown that the added sugar component of a food is greatly associated with its reinforcing value [8,9,27]. We have also shown that the highly reinforcing properties of sugar are difficult to overcome [10,11]. The activation of sweet taste receptors, the speed at which the information about a food is delivered from the chemosensory and somatosensory neurons in the mouth to the brain, and the magnitude of the activation of the food reward system govern the reinforcing and rewarding effect of sugar [28,29]. Low et al. [30] recently reported that the average concentration at which sugar can be differentiated from water is 9 mass percent (m%); however, interindividual variability is large (reported range from 2 m% to 32 m%). The sugar content of the chocolates provided in this study were 8 m%, 13 m%, 30 m%, and 48 m% for the 90% cocoa, 85% cocoa, 70% cocoa, and milk chocolates, respectively. Therefore, our finding that consuming the 85% cocoa, 70% cocoa, and milk chocolates resulted in significant increases in self-reported scores on the PEQ, and each subscale, above baseline is supported by the fact that all of these chocolates were above the 9 m% detection threshold.

In agreement with our prior study [6], consuming milk chocolate elicited a greater increase, compared to all the other chocolates, in the total number of positive responses as well as a greater increase in positive responses on the MBG subscale. The sugar content of the milk chocolate, which was equivalent to the upper sucrose detection threshold reported by Low et al. [30], may explain these results, as well as the mass appeal of milk chocolate. Indeed, participants in the current study consumed an average of 21 ± 25 g of milk chocolate compared to 6 ± 10 g of the 70% cocoa and 2 ± 4 g of the 85% cocoa chocolates. Taken together, these results agree substantively with other indicators of reinforcement essential for motivational behavior [15,31] and support the “addictive-like” behavior response individuals can experience with chocolate, in particular milk chocolate, consumption.

The finding that consuming chocolate containing 90% cocoa increased the total number of positive responses and the number of positive responses on the E subscale is of interest, given its reported “bitter” taste. The bioactive compounds found in dark chocolate may explain these positive results. In two double-blind, placebo control studies, Smit et al. [5] demonstrated that the amount of methylxanthines (theobromine and caffeine) found in dark chocolate can produce psychostimulant effects. They found that the consumption of both encapsulated cocoa powder and a “typical portion” of dark chocolate increases energetic arousal (“energetic”, “alert”, etc. versus “tired”, “sluggish”). This is consistent with our results demonstrating chocolate’s ability to increase physical and psychological feelings of excitement. Interestingly, when examined separately, the amount of theobromine found in a “typical portion” of chocolate (200–300 mg) does not appear to play a psychopharmacological role [32], whereas the amount of caffeine (25–35 mg) is well above the previously reported stimulatory threshold (12.5 mg [33]). The amount of caffeine consumed from the 90% cocoa chocolate sample in the current study was 9 mg, well below the stimulatory threshold. It may be that the theobromine found in chocolate enhances the stimulatory effect of caffeine. Thus, it is plausible that the combination of theobromine and caffeine found in chocolate provides an additive psychostimulant effect. While the methylxanthines in chocolate provide psychostimulant effects, Smit et al. [5], as well as Michener and Rozin [26], reported no difference in chocolate cravings between the ingestion of cocoa powder versus placebo, whereas consuming chocolate, including white chocolate (albeit to a lesser extent), immediately reduced chocolate cravings. As mentioned above, another explanation for this finding is that the sugar content of the 90% cocoa chocolate (8 m%) is well above the lowest and close to the average sucrose detection threshold previously reported [30]. Taken together, these results confirm that the combination of the main chocolate constituents is necessary to produce its psychoactive effects.

As expected, more milk chocolate was consumed than any of the other chocolates, indicating its reinforcing potential. This is consistent with the greater number of positive responses on the PEQ. However, contrary to our hypothesis, we did not find an association between the total amount of chocolate consumed or the total number of positive responses on the PEQ and BES scores. A potential explanation for this finding is all of our study participants had a score less than or equal to 17 and thus would be classified as non-binge eaters [20]. The narrow range in BES scores (2–17) of our participants may not have provided the heterogeneity needed to reliably determine a correlation between BES scores and chocolate consumption or self-reported PEQ scores. Further studies are needed to determine the potential correlation between BES scores and chocolate consumption or self-reported PEQ scores. Additionally, the low BES scores may also explain why we did not find an association between self-reported scores on the MBG subscale and chocolate consumption.

This study is not without limitations. The direct correlation between the percent cocoa and sugar content and the simultaneous changes of the fat content in the chocolate samples did not allow us to assess the independent psychoactive effects of each component. However, using commercially available chocolate provides a “real-world” value to our observations. Furthermore, tasting the chocolate samples sequentially in a relatively short amount of time may have produced an additive effect on our outcomes, such as habituation or sensory-specific satiety [34]. For this study, each participant consumed a total of 20 g of chocolate (5 g/sample) compared to the 12.5 g provided in our previous study [6] with similar PEQ results. Therefore, it does not appear that our within-subject study design had a significant impact on our results. Lastly, the participants in this study tended to be healthy-weight individuals (57% of the participants), thus, the results may not reflect the response of overweight and obese individuals. Although we did not find a significant effect of BMI on self-reported scores on the PEQ or any of the subscales, research has shown that food reinforcement can vary significantly between lean, overweight, and obese individual [35,36,37] and individuals with an eating disorder [38,39]. Further research is needed to determine if there exist differential psychoactive effects of chocolate consumption between lean, overweight, and obese populations and individuals with an eating disorder.

Our prior work suggests that obese individuals who engage in binge eating may do so because of an inherent reward deficiency, resulting in overeating high-sugar foods. We previously found an increase in food reinforcement after the consumption of a liquid chocolate-based preload, indicating a “sensitization” response to foods that increase the dopaminergic response [39]. Davis et al. [38] found that obese individuals diagnosed with binge eating disorder have a “hyper-reactivity to the hedonic properties of food, coupled with the motivation to engage in appetitive behaviors.” In women diagnosed with bulimia nervosa (BN), Bulik et al. [40] reported that food reinforcement decreases following deprivation. On the basis of these results, we posit that obese individuals with an eating disorder would demonstrate an overall increase in their self-reported scores on the PEQ due to reward sensitization. However, we would not anticipate a significant increase per se in obese individuals who do not exhibit a binge eating-related eating disorder.

Although questionnaires provide validated subjective measurements about food reward and eating behavior, the use of objective methodologies provide insight into the effect of highly palatable food on central dopamine activity and brain regions known to have a high density of dopamine neurons (positron emission tomography and functional magnetic resonance imaging, respectively). However, the cost of instrumentation, the expertise needed to operate the equipment and interpret the data, as well as the engineering and mechanical constraints of the scanners make objective measurements of the dopaminergic response to food challenging. A promising methodology, electroretinography (ERG), may provide a more efficient way to objectively assess central dopamine activity. ERG is a clinical ophthalmological procedure that overcomes the impediments of other neuroimaging methods. ERG records the electrical potential from the retina, which is dependent upon dopamine signaling [41,42], in response to light simulation. ERG has been used to show a negative correlation between self-reported cocaine craving and dopamine-mediated retinal signal [43,44] and a positive correlation between dopamine metabolite levels in the cerebrospinal fluid [45] and dopamine-mediated retinal signal. We have also used ERG to demonstrate a positive correlation between increased dopamine-mediated retinal signal with food stimulation and BES scores [46], consistent with positron emission tomography [47]. We are currently conducting research to extend those findings to chocolates varying in cocoa, sugar, and fat content. Results from this ongoing study should provide further evidence that ERG should be considered as a low-cost, non-invasive method for objective evaluation of the stimulating properties of food in conjunction with ARCI questionnaires.

## 5. Conclusions

Chocolate’s ability to modulate both the opioidergic and the dopaminergic systems, evident by the significant increase in self-reported scores on the PEQ, is consistent with research demonstrating chocolate’s reinforcing potential and comforting and mood-ameliorating effects [2,3,21,22,23]. These results help explain chocolate’s high reinforcing value, as foods that elevate feelings of euphoria, as indicated by an increase in the number of positive responses on the MBG subscales, have a greater reinforcing potential [17]. Given the measurable psychoactive dose–effect relationship observed in this study, we posit that chocolate’s ability to provoke “addictive-like” eating behavior is initiated by its bioactive compounds, and each incremental increase in added sugar further enhances these effects.

## Figures and Tables

**Figure 1 nutrients-11-00596-f001:**
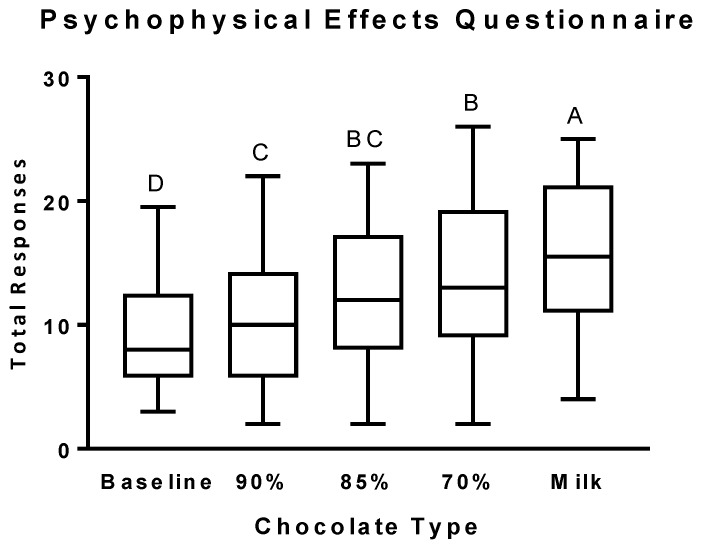
Total Psychophysical Effects Questionnaire (PEQ) scores after the consumption of chocolates differing in cocoa, sugar, and fat content. Self-reported scores on the PEQ questionnaire are presented as box and whiskers plots with the line representing the median, the box representing the 25th to 75th percentiles, and the whiskers representing the minimum to maximum values. Least-squares means are Baseline: 8.26 ± 1.16 (SE), 95% CI [5.93, 10.58]; 90% cocoa: 10.87 ± 1.10 (SE), 95% CI [8.65, 13.08]; 85% cocoa: 12.37 ± 1.10 (SE), 95% CI [10.15, 14.58]; 70% cocoa: 13.70 ± 1.10 (SE), 95% CI [11.49, 15.91]; Milk: 15.87 ± 1.10 (SE), 95% CI [13.65, 18.08]. Levels not connected by the same letter are significantly different.

**Figure 2 nutrients-11-00596-f002:**
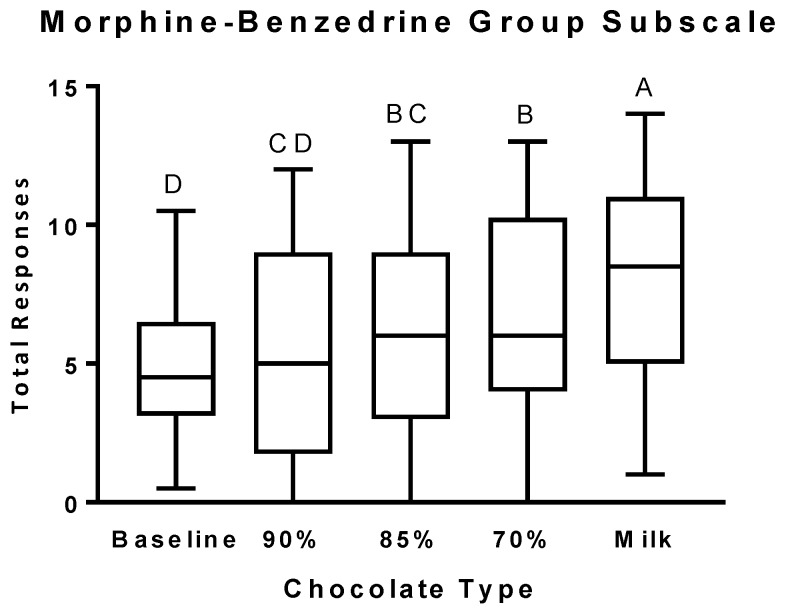
Morphine-Benzedrine Group (MBG) subscale scores after the consumption of chocolates differing in cocoa, sugar, and fat content. Self-reported scores on the MBG subscale are presented as box and whiskers plots with the line representing the median, the box representing the 25th to 75th percentiles, and the whiskers representing the minimum to maximum values. Least-squares means are Baseline: 4.28 ± 0.71 (SE), 95% CI [2.86, 5.69]; 90% cocoa: 5.37 ± 0.66 (SE), 95% CI [4.03, 6.70]; 85% cocoa: 6.10 ± 0.66 (SE), 95% CI [4.77, 7.43]; 70% cocoa: 6.63 ± 0.66 (SE), 95% CI [5.30, 7.97]; Milk: 7.97 ± 0.66 (SE), 95% CI [6.63, 9.30]. Levels not connected by the same letter are significantly different.

**Figure 3 nutrients-11-00596-f003:**
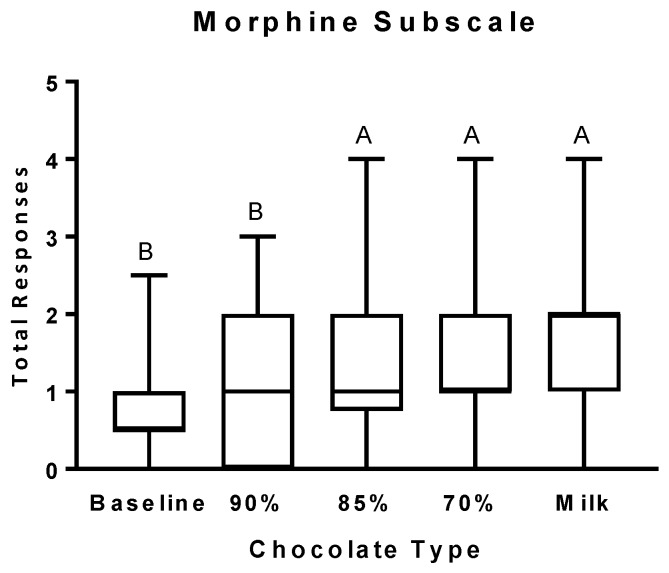
Morphine (M) subscale scores after the consumption of chocolates differing in cocoa, sugar, and fat content. Self-reported scores on the M subscale are presented as box and whiskers plots with the line representing the median, the box representing the 25th to 75th percentiles, and the whiskers representing the minimum to maximum values. Least-squares means are Baseline: 0.78 ± 0.20 (SE), 95% CI [0.37, 1.19]; 90% cocoa: 1.00 ± 0.19 (SE), 95% CI [0.62, 1.38]; 85% cocoa: 1.47 ± 0.19 (SE), 95% CI [1.09, 1.84]; 70% cocoa: 1.57 ± 0.19 (SE), 95% CI [1.19, 1.94]; Milk: 1.57 ± 0.19 (SE), 95% CI [1.19, 1.94]. Levels not connected by the same letter are significantly different.

**Figure 4 nutrients-11-00596-f004:**
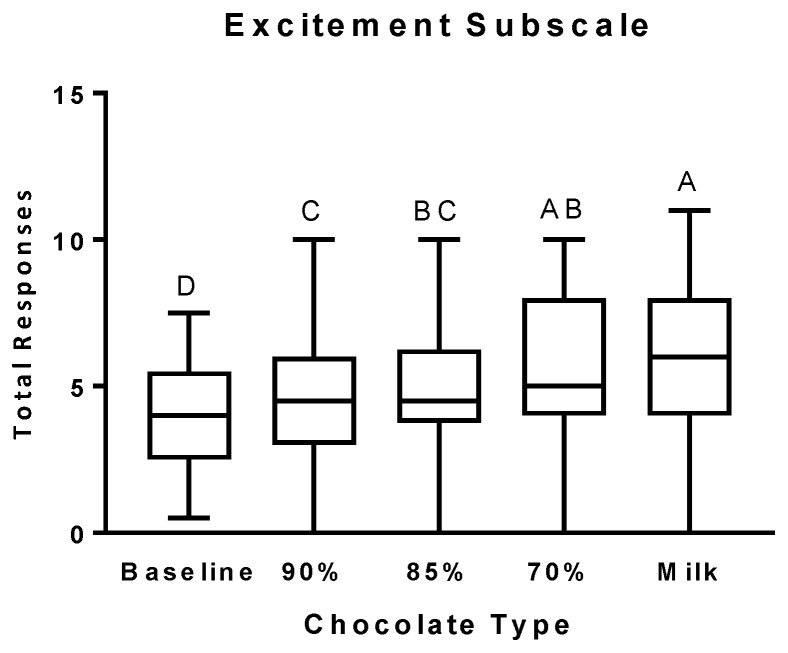
Excitement (E) subscale scores after the consumption of chocolates differing in cocoa, sugar, and fat content. Self-reported scores on the E subscale are presented as box and whiskers plots with the line representing the median, the box representing the 25th to 75th percentiles, and the whiskers representing the minimum to maximum values. Least-squares means are Baseline: 3.51 ± 0.47 (SE), 95% CI [2.56, 4.46]; 90% cocoa: 4.57 ± 0.44 (SE), 95% CI [3.68, 5.45]; 85% cocoa: 4.80 ± 0.44 (SE), 95% CI [3.92, 5.68]; 70% cocoa: 5.53 ± 0.44 (SE), 95% CI [4.65, 6.42]; Milk: 6.27 ± 0.44 (SE), 95% CI [5.38, 7.15]. Levels not connected by the same letter are significantly different.

**Table 1 nutrients-11-00596-t001:** Participant characteristics. BMI, body mass index.

	Grand Forks	Philadelphia	Total
N (F/M)	20 (14/6)	10 (6/4)	30 (20/10)
Age, years	24.1 ± 6.8	26.1 ± 6.0	24.8 ± 6.5
Height, cm	168.7 ± 9.3	172.3 ± 11.1	169.9 ± 9.9
Weight, kg	72.1 ± 10.9	70.3 ± 11.1	70.7 ± 9.4
BMI, kg/m^2^	25.3 ± 3.5	23.8 ± 4.3	24.8 ± 3.8

Values are means ± SD.

**Table 2 nutrients-11-00596-t002:** Characteristics of each 5 g chocolate sample.

Chocolate Type	kcal	Cocoa (%)	Sugar (g)	Fat (g)
Lindt^®^ milk	29	38	2.4	1.9
Lindt^®^ dark	30	70	1.5	2.4
Lindt^®^ extra dark	29	85	0.6	2.3
Lindt^®^ supreme dark	30	90	0.4	2.8

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
