# Peer review of "Increasing Chocolate’s Sugar Content Enhances Its Psychoactive Effects and Intake"

_nutrients, 2019, doi:10.3390/nu11030596_

Reviewer 1 Report

This study had a strong empirical design, and addressed a very important and currently relevant topic.  The results section, including figures and their captions, was especially clear.

 My minor suggestions are listed below:

-       Please add the BES to your abstract and discussion.

-       Line 40: I suggest starting the last sentence with “In a prior study, we…” (the sentence that cites reference #6) so readers know that you’re not referring to the current study.

-       In the introduction, it would help to have a brief discussion on the differences between opioid and dopamine activation in the brain, how these differentially impact reward and addiction, and why it is important.  Also, an explaination of why you chose to use these two specific pathways in your study (i.e., instead of oxytocin, serotonin, norepinephrine, etc.) would be useful to the reader.

-       The hypothesis for the BES at the end of the introduction is unclear.  You may want to explain it in a separate sentence.

-       Thank you for describing what each scale means on the ARCI MBG.  Since you explained that dopamine pathways are associated with the E scale, please state which scale the opioid pathways are associated with. 

-       The point about methylxanthines in the discussion was an insightful one.  I would suggest defining methylxanthines for readers who are not familiar with what that is.  

-       You may also consider discussing the possible role of caffeine content in the chocolate, and whether or not that could have affected the E scale. 

-       English/grammar edits are needed throughout.

 Great overall work!

Author Response

This study had a strong empirical design and addressed a very important and currently relevant topic.  The results section, including figures and their captions, was especially clear.

Thank you very much for the kind review. We believe that the manuscript was strengthened by your constructive comments.

My minor suggestions are listed below:

1)    Please add the BES to your abstract and discussion.

     We would like to think the reviewer for asking that we added the BES to the abstract and the discussion. In so doing, we discovered an error in the data. After reanalyzing with the corrected data there were no significant main effects of BES scores on chocolate consumption. We have added the BES to the abstract in both the methods (line 19) and results (lines 25-26) sections. In addition, we have added a new paragraph to the discussion (lines 299-309) to discuss the BES results. Below is the paragraph added to the discussion.

“As expected, more milk chocolate was consumed than any of the other chocolates, indicating its reinforcing potential. This is consistent with the greater number of positive responses on the PEQ. However, contrary to our hypothesis, we did not find an association between the total amount of chocolate consumed or the total number of positive responses on the PEQ and BES scores. A potential explanation for this finding is all of our study participants had a score less than or equal to 17, and thus would be classified as non-binge eaters [20]. The narrow range in BES scores (2 – 17) of our participants may not have provided the heterogeneity needed to reliably determine a correlation between BES scores and chocolate consumption or self-reported PEQ scores. Further studies are needed to determine the potential correlation between BES scores and chocolate consumption or self-reported PEQ scores. Additionally, the low BES scores may also explain why we did not find an association between self-reported scores on the MBG subscale and chocolate consumption.”         

2)    Line 40: I suggest starting the last sentence with “In a prior study, we…” (the sentence that cites reference #6) so readers know that you’re not referring to the current study.

             We have altered the manuscript as suggested.

3)    In the introduction, it would help to have a brief discussion on the differences between opioid and dopamine activation in the brain, how these differentially impact reward and addiction, and why it is important.  Also, an explanation of why you chose to use these two specific pathways in your study (i.e., instead of oxytocin, serotonin, norepinephrine, etc.) would be useful to the reader.

      Thank you for the suggestion. We have added a discussion (lines 51 – 58) about the differences between opioid and dopamine activation in the brain and why these two pathways are important to this research. Below is the additional discussion on this topic that was added to the introduction.

“Fat and sugar are known to stimulate both the dopamine and opioid neurotransmitter systems to regulate a food’s rewarding potential [12-14]. The dopamine neurotransmitter system stimulates ‘wanting’, or the motivation to consume the food [12,14], whereas the opioid neurotransmitter system modulates the consumption of a desired food by amplifying ‘liking’, or the hedonic value, of the desired food [12,13]. Thus, the sight, smell, and taste of a highly palatable food such as chocolate work together to trigger motivational and hedonic reward mechanisms that results in the pursuit and consumption of the desired food. Subjective dopaminergic and opioidergic effects of food consumption can be differentiated using the Addiction Research Center Inventory (ACRI) [15].”

4)    The hypothesis for the BES at the end of the introduction is unclear.  You may want to explain it in a separate sentence.

      We have added the hypothesis for the BES as a separate sentence (lines 78 – 80) for clarity. The BES hypothesis now reads:

“Furthermore, we hypothesized that BES scores would positively correlate with self-reported scores on the ARCI questionnaire and chocolate consumption.”

5)    Thank you for describing what each scale means on the ARCI MBG.  Since you explained that dopamine pathways are associated with the E scale, please state which scale the opioid pathways are associated with.

The MBG subscale contains questions that center on feelings of well-being and euphoria, which correlates with activation of both the dopaminergic and opioidergic systems. The M subscale focuses on attitude and physical sensations, which correlate with activation of the dopaminergic system. The E subscale relates to physical and psychological feelings of excitement, which correlate with activation of the dopaminergic system. We have added additional wording to clarify which neurotransmitter systems each scale corresponds to.

6)    The point about methylxanthines in the discussion was an insightful one.  I would suggest defining methylxanthines for readers who are not familiar with what that is. 

We have added theobromine and caffeine to line 280 to define methylxanthines for the reader.

7)    You may also consider discussing the possible role of caffeine content in the chocolate, and whether or not that could have affected the E scale.

The reviewer makes an excellent point about the caffeine content of the chocolate. We have added a discussion about potential effects of the caffeine in the chocolate on self-reported scores on the E scale (lines 2584 – 291). Below is the added discussion about the caffeine content of the chocolate.

“Interestingly, when examined separately, the amount of theobromine found in a “typical portion” of chocolate (200-300mg) does not appear to play a psychopharmacological role [32], whereas the amount of caffeine (25 - 35mg) is well above  the previously reported stimulatory threshold (12.5mg [33]). The amount of caffeine consumed from the 90% cocoa chocolate sample in the current study was 9mg, well below the stimulatory threshold. It may be that the theobromine found in chocolate enhances the stimulatory effect of the caffeine.[32] Thus, it is plausible that the combination of theobromine and caffeine found in chocolate provides an additive psychostimulant effect.” 

8)    English/grammar edits are needed throughout.

We have thoroughly gone through the manuscript to correct English/grammar errors.

Reviewer 2 Report

The study by Casperson et al. provides some conclusive evidence that there is a significant correlation between the sugar content of chocolate and its self-reported psychoactive effects in normal weight individuals. A second important conclusion is that the combination of all main chocolate ingredients, including cocoa, sugar and milk content, is necessary for chocolate to induce self-reported feelings of excitement and reward.

The above findings can have important ramifications for the understanding of the common psychoactive effects elicited by palatable food and substances of misuse and can lead to significant insights as to the impact of such stimuli on brain plasticity.  

There is relatively little to comment on the limitations of the study as the authors themselves comprehensively and thoughtfully describe the main ones, namely the lack of data on obese or ED patients and the exposure of participants to chocolate in a within design where each chocolate constituent was not independently manipulated to parse out individual ingredients' effects. However, the latter approach would not be necessarily productive as many types of chocolate to be tested under a more vigorous design would in principle not be palatable or represent a substantial departure from the average taste of commercially available chocolate. 

I would recommend to the authors to attempt to further discuss implications of the findings for obese and ED individuals. Would the PEQ and other score distributions shift to the left or to the right in these patients and how would that fit in terms of reward deficiency or reward sensitization as frequently discussed in the context of obesity and drug dependence?  I would also suggest that despite their use of validated and widely published questionnaires, the authors should share with the readers their thinking about how to obtain objective measures of psychoactive effects of chocolate in patients and how to analyze any such data in the context of their current findings. 

Minor: correct misspell on line 125 ("huger").

Author Response

The study by Casperson et al. provides some conclusive evidence that there is a significant correlation between the sugar content of chocolate and its self-reported psychoactive effects in normal weight individuals. A second important conclusion is that the combination of all main chocolate ingredients, including cocoa, sugar and milk content, is necessary for chocolate to induce self-reported feelings of excitement and reward.

The above findings can have important ramifications for the understanding of the common psychoactive effects elicited by palatable food and substances of misuse and can lead to significant insights as to the impact of such stimuli on brain plasticity.  

There is relatively little to comment on the limitations of the study as the authors themselves comprehensively and thoughtfully describe the main ones, namely the lack of data on obese or ED patients and the exposure of participants to chocolate in a within design where each chocolate constituent was not independently manipulated to parse out individual ingredients' effects. However, the latter approach would not be necessarily productive as many types of chocolate to be tested under a more vigorous design would in principle not be palatable or represent a substantial departure from the average taste of commercially available chocolate. 

1)    I would recommend to the authors to attempt to further discuss implications of the findings for obese and ED individuals.

We had added a new paragraph to the discussion (lines 327 – 336) to discuss the implications of the findings for obese and ED individuals. Below is the paragraph added to the discussion.

“Our prior work suggests that obese individuals who engage in binge eating may do so because of an inherent reward deficiency, resulting in overeating high-sugar foods. We previously found an increase in food reinforcement after the consumption of a liquid chocolate-based preload, indicating a “sensitization” response to foods that increase the dopaminergic response [39]. Davis et al. [38] found that obese individuals diagnosed with binge eating disorder have a “hyper-reactivity to the hedonic properties of food, coupled with the motivation to engage in appetitive behaviors.” In women diagnosed with bulimia nervosa (BN), Bulik et al. [40] reported that food reinforcement decreases following deprivation. Based on these results we posit that obese individuals with an eating disorder would demonstrate an overall increase in their self-reported scores on the PEQ due to reward sensitization. However, we would not anticipate a significant increase per se in obese individuals who do not exhibit a binge eating related eating disorder.”  

2)      Would the PEQ and other score distributions shift to the left or to the right in these patients

Based on prior research showing greater preference and motivation towards high sugar containing food, we would expect to see a rightward shift in a graft of increasing sugar content (decreasing cocoa content) verses PEQ results in those with binge related eating disorders. Our prior research demonstrated that obese individuals with a binge eating disorder have an increase in food reinforcement after consuming a liquid chocolate-based preload, indicating a “sensitization” response to foods that increase dopamine. This finding is supported by a recent publication by Goodman, et al. (Eat Behav. 2018 Jan;28:8-15. doi: 10.1016/j.eatbeh.2017.11.005. Epub 2017 Nov 17). We might also expect a rightward shift in response in those with Bulimia Nervosa, based on Bulik, et al. (Physiol Behav. 1994 Apr;55(4):665-72.). However, we would not expect a rightward shift per se in those who are obesity, who do not exhibit a binge eating related eating disorder. These details were added to the discussion (lines 334 – 336).

3)      How would that fit in terms of reward deficiency or reward sensitization as frequently discussed in the context of obesity and drug dependence?  

        As mentioned above, those who are obese with Binge Eating Disorder experience a sensitization response to the reward of high sugar/high fat food.

4)      I would also suggest that despite their use of validated and widely published questionnaires, the authors should share with the readers their thinking about how to obtain objective measures of psychoactive effects of chocolate in patients and how to analyze any such data in the context of their current findings. 

We published a study using electroretinography (ERG) (Nasser et al 2013, Obesity 21:976-980), that demonstrated that the ERG response to light, which is dependent upon dopamine release, was increased by tasting chocolate to an equivalent amount as a 20mg oral dose of methylphenidate, a dopamine agonist drug. We are currently conducting research to extend those findings to chocolate varying in cocoa, sugar and fat content. Results from that study should provide a method for objective evaluation of psychoactive effect of chocolate. We can’t comment further until the experiment is concluded. We have added a paragraph to the discussion (lines 337 – 355) about objective measures to determine the dopaminergic response to highly-reinforcing foods. Below is the paragraph that was added to the discussion.

“Although questionnaires provide validated subjective measurements about food reward and eating behavior, the use of objective methodologies (e.g., positron emission tomography, functional magnetic resonance imaging) provide insight into the effect of highly palatable food on central dopamine activity and brain regions known to have high density of dopamine neurons (PET and fMRI respectively). However, the cost of instrumentation, the expertise needed to operate the equipment and interpret the data, as well as, engineering and mechanical constraints of the scanners make objective measurements of the dopaminergic response to food challenging. A promising methodology, electroretinography (ERG), may provide a more efficient way to objectively assess central dopamine activity. ERG is a clinical ophthalmological procedure that overcomes the impediments of other neuroimaging methods. ERG records the electrical potential from the retina, which is dependent upon dopamine signaling [41,42], in response to light simulation. ERG has been used to show a negative correlation between self-reported cocaine craving [43,44] and positive correlation between dopamine metabolite levels in cerebrospinal fluid [45] and dopamine-mediated retinal signal. We have also used ERG to demonstrate a positive correlation between increased dopamine-mediated retinal signal with food stimulation and BES scores [46]; consistent with positron emission tomography [47]. We are currently conducting research to extend those findings to chocolate varying in cocoa, sugar, and fat content. Results from that study should provide further evidence that ERG should be considered as a low-cost, non-invasive method for objective evaluation of the stimulating properties of food in conjunction with ARCI questionnaires.”

5)      Minor: correct misspell on line 125 ("huger").

Thank you for pointing this misspelling out to us. The misspelling has been corrected.